# (-)-α-Pinene reduces quorum sensing and *Campylobacter jejuni* colonization in broiler chickens

Katarina Šimunović[1], Orhan Sahin[2], Jasna Kovač[3], Zhangqi Shen[2¤], Anja Klančnik[1], Qijing Zhang[2], Sonja Smole Možina[1]*

**1** Department of Food Science and Technology, Biotechnical Faculty, University of Ljubljana, Ljubljana, Slovenia, **2** College of Veterinary Medicine, Iowa State University, Ames, Iowa, United States of America, **3** Department of Food Science, The Pennsylvania State University, University Park, Pennsylvania, United States of America

¤ Current address: Beijing Advanced Innovation Centre for Food Nutrition and Human Health, College of Veterinary Medicine, China Agricultural University, Beijing, China.

* sonja.smole-mozina@bf.uni-lj.si

**Citation:** Šimunović K, Sahin O, Kovač J, Shen Z, Klančnik A, Zhang Q, et al. (2020) (-)-α-Pinene reduces quorum sensing and *Campylobacter jejuni* colonization in broiler chickens. PLoS ONE 15(4): e0230423. https://doi.org/10.1371/journal.pone.0230423

**Data Availability Statement:** "All data for the manuscript is available in Figshare, specifically: Fig 1. 10.6084/m9.figshare.8234153 https://figshare.com/s/23ec3f570d93069ae7fb Fig 2. 10.6084/m9.

## Abstract

*Campylobacter jejuni* is one of the most prevalent causes of bacterial gastroenteritis worldwide, and it is largely associated with consumption of contaminated poultry. Current *Campylobacter* control measures at the poultry production level remain insufficient, and hence there is the need for alternative control strategies. We evaluated the potential of the monoterpene (-)-α-pinene for control of *C. jejuni* in poultry. The antibacterial and resistance-modulatory activities of (-)-α-pinene were also determined against 57 *C. jejuni* strains. In addition, the anti-quorum-sensing activity of (-)-α-pinene against *C. jejuni* NCTC 11168 was determined for three subinhibitory concentrations (125, 62.5, 31.25 mg/L) over three incubation times using an autoinducer-2 bioassay based on *Vibrio harveyi* BB170 bioluminescence measurements. The effects of a subinhibitory concentration of (-)-α-pinene (250 mg/L) on survival of *C. jejuni*, and in combination with enrofloxacin on fluoroquinolone resistance development in *C. jejuni*, were determined in a broiler chicken model, by addition of (-)-α-pinene to the broiler water supply. The reduction of *C. jejuni* numbers by (-)-α-pinene was further determined in broiler chickens that were colonized with either fluoroquinolone-susceptible or -resistant strains, by direct gavage treatment. We observed weak *in vitro* antimicrobial activity for (-)-α-pinene alone (MIC >500 mg/L), but strong potentiating effects on antibiotics erythromycin and ciprofloxacin against different *Campylobacter* strains (>512 fold change). After 24 h of treatment of *C. jejuni* with (-)-α-pinene, its quorum-sensing signaling was reduced by >80% compared to the untreated control. When given in the drinking water, (-)-α-pinene did not show any significant inhibitory effects on the level of *C. jejuni* in the colonized chickens, and did not reduce fluoroquinolone resistance development in combination with enrofloxacin. Conversely, when (-)-α-pinene was administered by direct gavage, it significantly reduced the number of fluoroquinolone susceptible *C. jejuni* in the colonized broiler chickens. These results demonstrate that (-)-α-pinene modulates quorum-sensing in *Campylobacter*,

figshare.8234192 https://figshare.com/s/
aa309738e3cdbe99717e Fig 3. 10.6084/m9.
figshare.8234189 https://figshare.com/s/
55589de9346a17749cd5 S1 and S2 Figs 10.6084/
m9.figshare.8234156 https://figshare.com/s/
2559324ea098365d3e8c S1 and S2 Tables. 10.
6084/m9.figshare.8234159 https://figshare.com/s/
9a452dc1b939e7d9de30".

**Funding:** This study was financed by the
Slovenian-American bilateral projects BI-SLO-USA
2014/2015 and 2018/19 and P4-0116 funded by
the Slovenian Research Agency (ARRS) awarded
to SSM. http://www.arrs.si/en/index.asp JK was
supported by the USDA National Institute of Food
and Hatch Appropriations under Project
#PEN04646 and Accession #1015787. https://nifa.
usda.gov/ The funders had no role in study design,
data collection and analysis, decision to publish, or
preparation of the manuscript.

**Competing interests:** The authors have declared
that no competing interests exist.

potentiates antibiotics against different *Campylobacter* strains, and reduces *Campylobacter* colonization in broiler chickens.

## Introduction

*Campylobacter jejuni* represents a food safety hazard worldwide. It can cause campylobacteriosis, which is one of the most widespread bacterial foodborne zoonoses reported for the European Union and the United States [1–3]. Campylobacteriosis is commonly associated with ingestion of contaminated poultry, water, or milk, and manifests as acute watery/bloody diarrhea, fever, and cramps. This can also lead to post-infection development of the severe neurological condition known as Guillain-Barre syndrome [1]. An additional risk is the increasing antimicrobial resistance in *Campylobacter*. In particular, *Campylobacter* resistance to fluoroquinolones and macrolides compromises effectiveness of antibiotic therapies and poses a heightened food safety concern in the food chain [2,4].

Currently, there are no fully effective and practical measures for the control of poultry contamination with the avian commensal *C. jejuni*. Control of *Campylobacter* includes pre-harvest measures on poultry farms and post-harvest approaches in processing plants. Pre-harvest biosecurity and hygiene measures can be used to prevent entrance of *Campylobacter* onto a farm and to limit its spread between flocks, whereas post-harvest measures focus on decontamination of carcasses [5–7].

To mitigate transmission of *Campylobacter* from food animals to humans through the food supply chain, effective pathogen control measures are needed. These must be designed to reduce the *Campylobacter* load at the farm and/or slaughterhouse level, with emphasis on poultry production, where *Campylobacter* resides as a commensal [2,4,8]. Even a relatively small reduction in *C. jejuni* numbers in the chicken cecum by 1 $\log_{10}$ CFU can reduce the public health risk by more than 50% [8].

A number of natural products have been shown to have anti-*Campylobacter* activities and have been studied as feed additives, such as essential oils and their components [9,10]. The majority of studies have been focused on the bactericidal aspects of the antimicrobial actions of natural compounds, while their potential for reduction of pathogen virulence through inhibition of efflux pumps, quorum sensing or other factors contributing to colonization of a host, remains largely unexplored [11,12].

In *C. jejuni*, quorum sensing is mediated by the furanosyl borate diester autoinducer-2 (AI-2) signal that is produced as a result of the action of the *S*-ribosylhomociateinase LuxS, encoded by the *luxS* gene [13]. The *C. jejuni* mutant lacking the *luxS* gene shows impaired biofilm formation, motility, resistance against oxidative stress, invasion of Caco-2 cells, virulence in the host, and colonization of the chicken intestine [13–18]. This suggests that inhibition of *C. jejuni* quorum sensing in the host might result in reduction of *C. jejuni* in the feces, and thus control *C. jejuni* spread in the environment.

Only a few plant extracts have been reported to show anti-quorum-sensing effects in *C. jejuni* (e.g., citrus extracts, *Evodia ruticarpa* extracts) to date [19,20].

In a previous study, we also demonstrated efflux-inhibitory and resistance-modulatory activities of the monoterpene (-)-α-pinene in *Campylobacter* [21]. These findings suggest that plant extracts, such as (-)-α-pinene, modulate multiple physiological functions in *C. jejuni*. However, the effects of these plant extracts have not been examined using an *in vivo* system, which would allow for determination of their potential use in food-animal production. In this

study we further investigated (-)-α-pinene bioactivities, including: (i) inhibition of *C. jejuni* quorum sensing *in vitro*; (ii) modulation of *C. jejuni* resistance to fluoroquinolones in broiler chickens; and (iii) reduction of *C. jejuni* colonization in broiler chickens.

## Materials and methods

### Bacterial strains and growth conditions

The *Campylobacter jejuni* strains shown in S1 and S2 Tables were isolated and characterized by Luangtongkum et al. [22], and were stored at -80˚C in 80% Mueller Hinton broth (MHB: Oxoid, UK) with 20% glycerol. They were then grown on Mueller-Hinton agar (MHA; Oxoid, UK) at 42˚C under microaerobic conditions (5% $O_2$, 10% $CO_2$, 85% $N_2$) for 24 h. The second passage from each culture was used in the experiments. When necessary, MHA was supplemented with selective medium (SR01176; Oxoid, UK) and growth medium (SR0232E; Oxoid, UK) (MHA-SS), 30 mg/L kanamycin (Merck, Germany), or 4 mg/L ciprofloxacin (Merck, Germany). The *Vibrio harveyi* BB170 reporter strain [19,23] was grown on autoinducer bioassay (AB) medium at 30˚C, which contained 17 g/L NaCl (Merck, Germany), 12.3 g/L $MgSO_4$ (Merck, Germany), 2 g/L casamino acids (BD Bacto; Fisher Scientific), 1 mM $K_2HPO_4$ (Kemika, Croatia), 0.1 mM L-arginine (Sigma Aldrich, Germany), and 1% (v/v) glycerol (Kemika, Croatia).

### Antimicrobial and resistance-modulatory activities of (-)-α-pinene *in vitro*

The minimal inhibitory concentrations (MICs) of (-)-α-pinene (Sigma Aldrich, Germany) were determined against all of the 57 *C. jejuni* strains that were sourced according to S1 Table, using the broth microdilution method, as described previously [21]. The reported $MIC_{50}$ and $MIC_{90}$ values represent the MICs that inhibited at least 50% and 90%, respectively, of the tested strains. The resistance-modulatory activity of (-)-α-pinene was determined in combination with the clinically relevant antibiotics ciprofloxacin and erythromycin (Fluka Chemie, Germany), using the broth microdilution method [21]. (-)-α-Pinene was added to these antibiotics at the subinhibitory concentration of 125 mg/L. The MICs were determined, along with the fold-changes (FC) between the MICs of the antibiotics alone and their MICs with the addition of (-)-α-pinene. These were calculated according to Eq (1):

$$FC = MIC_{Ab}/MIC_{AbAp}, \tag{1}$$

where $MIC_{Ab}$ is the MIC of the antibiotic alone, and $MIC_{AbAp}$ is the MIC of the antibiotic in the presence of 125 mg/L (-)-α-pinene. FC $\geq$2 was considered as indicative of biologically significant resistance modulation.

### Quorum-sensing inhibition *in vitro*

To determine the influence of (-)-α-pinene on *C. jejuni* quorum sensing, autoinducer-2 bioassays were performed. *C. jejuni* NCTC 11168 and *C. jejuni* 11168Δ*luxS* (negative control; [18]) cultures in MHB were adjusted to $OD_{600}$ 0.1. The (-)-α-pinene stock solutions were prepared in 100% dimethylsulfoxide (DMSO) at 6.25 g/L, 12 g/L, and 25 g/L. Fifty microliters of each stock was added to 10 mL of each culture for the final (-)-α-pinene concentrations of 31.25 mg/L, 62.5 mg/L, and 125 mg/L. Untreated cultures were used as controls. The cultures were incubated under microaerobic conditions at 42˚C for 24 h. Samples of 3 mL were taken after 4 h, 8 h, and 24 h, and filter sterilized using 0.2-μm syringe filters (Sartorius, Germany), for the cell-free supernatants.

The autoinducer-2 bioassay was performed as previously described [19], with some modifications. The quorum-sensing inhibition bioassays were carried out using a *V. harveyi* BB170 reporter strain [23] that was grown for 16 h at 30˚C and 150 rpm, and used at the final concentration of $5 \times 10^4$ CFU/mL in AB medium. Filter sterilized *C. jejuni* cell-free supernatants were added to the suspensions of the reporter strain to a final concentration of 10% (v/v) (i.e., 20 μL cell-free supernatant added to 180 μL reporter strain suspension). Sterile medium was used as the blank (10% [v/v] MHB, 90% [v/v] AB medium). Kinetic measurements were carried out for the bioluminescence signals of *V. harveyi* BB170 produced as a result of the presence of the quorum-sensing signal that originated from the *C. jejuni* cell-free supernatants. The relative luminescence signals were measured at 15-min intervals over 20 h at 30˚C, in white microtiter plates (Nunc, Thermo Scientific) incubated in a microplate reader (Varioskan Lux; Thermo Scientific).

*Vibrio harveyi* produces a background luminescence signal that increases with the concentration of the culture. To define the most stable point of signal production, *V. harveyi* growth and signal production was measured in AB supplemented with MHB (180 μL:20 μL) at 30˚C. The signal stabilized when *V. harveyi* entered the stationary phase (S1 Fig). The time point when *V. harveyi* enters the stationary phase (after 9 h incubation) was used in the calculation of the quorum-sensing signals attributed to *C. jejuni*.

The relative luminescence signals were interpreted as the quorum-sensing signal in the *C. jejuni* cell-free supernatants (i.e., a higher signal indicated a higher concentration of quorum-sensing signaling molecules produced by *C. jejuni*), and are shown in S2 Fig.

Cell-free supernatants from *C. jejuni* 11168Δ*luxS*, a mutant that cannot produce the quorum-sensing signal (AI-2), were used as the negative control, and fresh MHB as the blank. To determine the inhibition rates of the quorum sensing by (-)-α-pinene, the blank values were subtracted from all of the test sample values. These corrected test values were used to calculate the reduction in quorum sensing using Eq (2):

$$\text{Quorum−sensing inhibition (\%)} = 100 − ((C.\ jejuni \text{ treated with } (−)−α−\text{pinene}/\text{untreated } C.\ jejuni) \times 100). \quad (2)$$

The experiments were performed as three independent biological replicates and three technical replicates.

## Broiler chicken colonization with *C. jejuni*

Broiler chicks (Cornish Rock strain, unspecified sex) were obtained from the Welp Hatchery in Iowa (USA) on the day of hatching, and were divided into four groups of 10 broilers each. The broiler chickens were kept in sanitized wire-floored cages (each group, n = 10/cage), and provided with feed and water *ad libitum*. Cloacal swabs were taken from each broiler chicken prior to the experiment and plated onto MHA-SS to confirm that they showed no *Campylobacter* colonization prior to inoculation. No *Campylobacter* was detected in any of the broiler chickens tested. At the age of day 5, each bird was inoculated with $3.6 \times 10^6$ CFU *C. jejuni* NCTC 11168 by oral gavage. To confirm colonization, cloacal swabs were collected 3 days after the inoculation.

At the age of day 8, medicated water was given to birds for 5 consecutive days to evaluate the synergistic effects of (-)-α-pinene and enrofloxacin on *Campylobacter* fluoroquinolone resistance development. Since enrofloxacin and (-)-α-pinene were dissolved in DMSO, the medicated water contained 0.5% DMSO for all groups, with the following additions for each group: (1) none (DMSO; control group); (2) 250 mg/L (-)-α-pinene (AP); (3) 50 mg/L enrofloxacin (ENRO) (Sigma Aldrich); and (4) 250 mg/L (-)-α-pinene and 50 mg/L enrofloxacin

(ENRO+AP). Cloacal swabs were collected every other day, and 3 days after (day 16 of age) the final day of the treatment for *Campylobacter* culture.

As the culture results of the cloacal swabs showed, all of the birds in all of the groups were colonized by *C. jejuni*. In addition, enrofloxacin treatment resulted in development of fluoro-quinolone resistance in *C. jejuni* in the treated groups (FQ-R; groups 3 (ENRO) and 4 (ENRO +AP) above), while the groups that were not treated with enrofloxacin remained colonized by fluoroquinolone sensitive *C. jejuni* (FQ-S; groups 1 (DMSO) and 2 (AP) above). To further determine the effects of (-)-α-pinene on susceptible and resistant *C. jejuni in vivo*, one group of each category (groups 2 and 4) were given an additional 250 mg/L (-)-α-pinene directly by oral gavage (i.e., the FQ-S treated and FQ-R treated groups) while the other two (groups 1 and 3) did not receive any (-)-α-pinene (i.e., the FQ-S untreated and FQ-R untreated groups). The gavage water (0.4 mL/bird/day) was started at the age of 18 days for 3 consecutive days, and it contained 0.5% DMSO for all four groups. Direct gavage treatment was used to minimize the variability of the dosing between the broiler chickens. All of the broiler chickens were sacrificed at 21 days of age, at which time cecum contents were collected for *Campylobacter* culture.

For determination of *Campylobacter* numbers, all of the fecal swabs and the cecum contents collected were suspended in MHB (1 mL MHB/swab with 100 mg feces), serially diluted, plated onto MHA-SS (for total *C. jejuni* numbers) and onto MHA-SS supplemented with 4 mg/L cip-rofloxacin (for fluoroquinolone-resistant *C. jejuni*), and incubated at 42˚C under microaerobic conditions for 48 h. The detection limit of the culture method for *C. jejuni* was 100 CFU/g feces. To further confirm the emergence of fluoroquinolone-resistant *C. jejuni* mutants, colonies from MHA-SS were also collected for each group at every sampling, and antimicrobial sensitivity test-ing was performed using E-test strips (0.002–32 mg/L ciprofloxacin; AB Biodisk, Sweden).

### Ethics statement

All of the animal protocols and procedures used in this study were reviewed and approved by the Institutional Animal Care and Use Committee (IACUC) at Iowa State University (Ames, Iowa, USA) before the start of the experiments. The approved protocol identification number is: 2-07-6304-G. The animal care and use protocol used in this study adhered to regulations and guide-lines provided in the "Guide for the Care and Use of Laboratory Animals", 8[th] edition, and the "Guide for the Care and Use of Agricultural Animals in Research and Teaching", 3[rd] edition.

### Statistical analyses

All of the data were tested for normality with Kolmogorov-Smirnov and Shapiro-Wilk tests. The statistical significances of the quorum-sensing inhibition and antimicrobial and resis-tance-modulatory activities were calculated using one-way ANOVA with Tukey's *post-hoc* tests. The associations between antibiotic resistance and resistance modulation were calculated using Chi-squared tests with Cramer's V strength tests. Differences in colonization between the treated and untreated broiler chickens were analyzed using Student's t-tests. Emergence of fluoroquinolone-resistant mutants in groups was compared using Student's t-tests. All of the analyses were performed using the SPSS software, version 21 (IBM Corp., Armonk, NY, USA).

## Results

### (-)-α-Pinene shows weak antimicrobial activity but strong resistance-modulatory activity against *C. jejuni in vitro*

To evaluate the clinical relevance of previously reported antimicrobial and resistance-modula-tory activities of (-)-α-pinene [21], these activities were tested across 57 broiler, turkey, and

human *C. jejuni* strains, in addition to the reference strain (NCTC 11168), which were sourced as listed in S1 Table. The following criteria were defined for the antimicrobial activities of (-)-α-pinene alone: high: MIC ≤31.25 mg/L; intermediate: MIC from 62.5 mg/L to 1000 mg/L; low: MIC at 2000 mg/L; none: MIC >2000 mg/L. Based on these criteria, and collectively considering these 57 *C. jejuni* strains, (-)-α-pinene alone showed low antimicrobial activity, with the overall $MIC_{50}$ of 2000 mg/L (concentration of (-)-α-pinene that inhibited at least 50% of the strains; Table 1). Considering the strains individually, the majority of these strains showed low antimicrobial activities of (-)-α-pinene (n = 39; 68%), with no effects seen against 12% (n = 7) (Table 1). These data thus demonstrate the relatively weak antimicrobial activity of (-)-α-pinene alone against *C. jejuni*.

The resistance-modulatory activity of (-)-α-pinene in *C. jejuni* with two clinically important antibiotics (i.e., ciprofloxacin, erythromycin) was tested at the subinhibitory concentration of 125 mg/L (-)-α-pinene. These data are reported as fold-changes (FC) in terms of the decrease in the MICs of the antibiotics when combined with (-)-α-pinene (Table 1). The following criteria were set for the resistance-modulatory activities in terms of the fold-changes: high: ≥32; intermediate: <32 to ≥8; low: <8 to ≥2; and no activity, 1.

The FC differed among the strains, from 1 (i.e., no activity) to >512 (i.e., high activity). When combined with ciprofloxacin, (-)-α-pinene showed strong and intermediate resistance-modulatory activities in 39% (n = 22) and 18% (n = 10) of the strains, respectively. The susceptibility to ciprofloxacin was affected marginally by (-)-α-pinene (i.e., low activity) in 37% (n = 21) of the strains, and not affected at all in 7% (n = 4) of the strains. The antimicrobial activity of erythromycin was enhanced by (-)-α-pinene in the majority of the tested strains. In 46% (n = 26), (-)-α-pinene showed high resistance-modulatory activity; in 13% (n = 7), intermediate, and in 32% (n = 18), low activity. (-)-α-Pinene did not increase the susceptibility to erythromycin in 9% (n = 5) of the tested strains. It was interesting to note that only strains M33323 and W14861 did not show any changes in susceptibility to both ciprofloxacin and erythromycin when combined with (-)-α-pinene.

We then compared the resistance-modulatory activity of (-)-α-pinene for ciprofloxacin and erythromycin with the available antibiotic susceptibility data for a range of antibiotics (i.e., ampicillin, tetracycline, kanamycin, gentamicin, erythromycin, clindamycin, ciprofloxacin, nalidixic acid, norfloxacin), using 37 of the broiler and turkey strains (S2 Table). Here, no significant associations were seen between the susceptibilities to any specific antibiotic and the resistance-modulatory activities of (-)-α-pinene. Thus, these resistance-modulating activities of (-)-α-pinene did not depend on the susceptibility to any of the antibiotics tested.

### *Campylobacter jejuni* quorum sensing is inhibited by (-)-α-pinene *in vitro*

To determine the potential of (-)-α-pinene for inhibition of quorum sensing, *C. jejuni* NCTC 11168 was treated with three subinhibitory concentrations of (-)-α-pinene (i.e., 31.25, 62.5, 125 mg/L) for 4 h, 8 h, and 24 h. The reductions in the quorum-sensing signaling molecules produced in the treated cultures were calculated and compared to that for the untreated cultures. Inhibition of *C. jejuni* quorum sensing was seen for all of these samples treated with (-)-α-pinene, regardless of the concentration added, and at all time-points (Fig 1). After 8 h treatment with (-)-α-pinene, the quorum-sensing inhibition was in the same range for all of the (-)-α-pinene concentrations used (10%-13% inhibition; $p$ >0.05). After 4 h and 8 h of treatments, the highest quorum-sensing inhibition by (-)-α-pinene did not exceed 20%. After 24 h of treatment, there was higher quorum-sensing inhibition in all of the samples treated with (-)-α-pinene, compared to the shorter incubation times ($p$ <0.01). The 24-h treatment with 31.25 mg/L (-)-α-pinene resulted in 36% inhibition of quorum sensing, while the higher

**Table 1. Antimicrobial and resistance modulatory activity of (-)-α-pinene with antibiotics ciprofloxacin and erythromycin in 57 *Campylobacter jejuni* strains from chicken meat (strain code, CB), turkey meat (strain code, CT), human feces (strain codes F, X) and the reference strain NCTC 11168.**

| Strain code | (-)-α-Pinene MIC (mg/L) | Ciprofloxacin MIC (mg/L) | | Fold | Erythromycin MIC (mg/L) | | Fold |
|---|---|---|---|---|---|---|---|
| | Alone | Alone | Plus (-)-α-pinene[a] | change[b] | Alone | Plus (-)-α-pinene[a] | change[b] |
| CB1:6 | 1000 | 16 | <0.125 | >128 | 0.5 | <0.002 | >256 |
| CB1:14 | 1000 | 16 | <0.125 | >128 | 0.5 | <0.002 | >256 |
| CB1:18 | 1000 | 16 | <0.125 | >128 | 0.5 | 0.125 | 4 |
| CB2:6 | 2000 | 64 | 16 | 4 | 0.5 | 0.06 | 8 |
| CB2:8 | 2000 | 64 | 32 | 2 | 0.5 | 0.25 | 2 |
| CB2:11 | 1000 | 8 | 4 | 2 | 0.5 | 0.25 | 2 |
| CB3:1 | 2000 | 0.125 | 0.06 | 2 | 0.25 | 0.06 | 4 |
| CB3:5 | 2000 | 0.25 | 0.06 | 4 | 0.5 | 0.25 | 2 |
| CB 4:21 | 1000 | 0.06 | 0.03 | 2 | 0.06 | <0.002 | >32 |
| CB 4:22 | 2000 | 0.125 | 0.001 | 128 | 0.06 | 0.002 | 32 |
| CB 6:8 | 2000 | 0.06 | 0.008 | 8 | 0.125 | <0.002 | >64 |
| CB 6:9 | 2000 | 0.06 | 0.03 | 2 | 0.125 | 0.06 | 2 |
| CB 6:26 | 2000 | 0.06 | 0.03 | 2 | 0.25 | 0.125 | 2 |
| CB 7:15 | 1000 | 8 | <0.06 | >128 | 0.25 | <0.002 | >128 |
| CB 7:21 | 2000 | 8 | 1 | 8 | 0.125 | <0.002 | >64 |
| CB 8:14 | 2000 | 0.06 | 0.001 | 64 | 0.125 | 0.002 | 64 |
| CB 8:15 | 1000 | 0.06 | 0.001 | 64 | 0.5 | <0.002 | >256 |
| CT 1:1 | 2000 | 16 | 1 | 16 | 0.5 | 0.03 | 16 |
| CT 1:9 | 1000 | 16 | 2 | 8 | 0.5 | 0.06 | 8 |
| CT 2:2 | 2000 | 16 | <0.06 | >256 | 256 | <1 | >256 |
| CT 3:5 | 2000 | 0.06 | <0.001 | >64 | 0.03 | <0.002 | 16 |
| CT3:11 | 500 | 4 | <0.06 | >64 | - | - | - |
| CT3:19 | 2000 | 16 | <0.06 | >256 | 512 | <1 | >512 |
| CT4:4 | 2000 | 16 | 8 | 2 | 128 | 64 | 2 |
| CT4:14 | 2000 | 8 | 8 | 1 | 128 | 64 | 2 |
| CT5:2 | 2000 | 16 | 4 | 4 | 256 | 256 | 1 |
| CT5:8 | 2000 | 8 | 2 | 4 | 256 | 16 | 16 |
| CT5:10 | 2000 | 16 | <0.06 | >256 | 256 | <1 | >256 |
| CT5:12 | 2000 | 16 | 4 | 4 | 256 | 32 | 8 |
| CT5:18 | 2000 | 16 | 4 | 4 | 256 | 64 | 4 |
| CT 6:18 | 2000 | 8 | 0.5 | 16 | 128 | <1 | >128 |
| CT 6:8 | 2000 | 16 | 2 | 8 | 256 | <1 | >256 |
| CT 6:16 | 2000 | 0.03 | <0.001 | >32 | 128 | <1 | >128 |
| CT 7:2 | 2000 | 0.03 | <0.001 | >32 | 0.06 | 0.002 | 32 |
| CT 8: 28 | 2000 | 8 | 1 | 8 | 0.25 | <0.008 | >32 |
| CT 8:29 | >2000 | 4 | <0.06 | >64 | 64 | 8 | 8 |
| CT 8:22 | 2000 | 4 | 0.5 | 8 | 0.5 | 0.5 | 1 |
| CT 9:14 | 2000 | 0.25 | <0.002 | >128 | 2 | <0.008 | >256 |
| CT 10:18 | 1000 | 0.06 | <0.002 | >32 | 0.25 | <0.008 | >32 |
| CT 9:21 | 1000 | 0.125 | <0.002 | >64 | 2 | <0.008 | >256 |
| F6501 | >2000 | 0.125 | 0.06 | 2 | 0.25 | 0.125 | 2 |
| H2958 | 2000 | 0.125 | 0.03 | 4 | 0.5 | 0.25 | 2 |
| M63885 | 2000 | 0.25 | <0.002 | >128 | 0.5 | <0.008 | >64 |
| T59822 | 2000 | 0.125 | 0.016 | 8 | 0.25 | 0.06 | 4 |

(*Continued*)

**Table 1.** (Continued)

| Strain | (-)-α-Pinene | Ciprofloxacin | | | Erythromycin | | |
|---|---|---|---|---|---|---|---|
| code | MIC (mg/L) | MIC (mg/L) | | Fold | MIC (mg/L) | | Fold |
| | Alone | Alone | Plus (-)-α-pinene[a] | change[b] | Alone | Plus (-)-α-pinene[a] | change[b] |
| W14861 | >2000 | 0.25 | 0.25 | 1 | 2 | 2 | 1 |
| X60179 | 2000 | 8 | <0.06 | >128 | 0.25 | <0.002 | >128 |
| F15871 | >2000 | 0.125 | 0.06 | 2 | 1 | 0.5 | 2 |
| W11805 | 2000 | 0.06 | 0.031 | 2 | 0.25 | 0.06 | 4 |
| M402 | >2000 | 0.06 | 0.031 | 2 | 1 | 0.5 | 2 |
| W28752 | 2000 | 0.06 | 0.008 | 8 | 0.5 | <0.002 | >256 |
| M33323 | 2000 | 0.06 | 0.06 | 1 | 0.25 | 0.25 | 1 |
| W64861 | 2000 | 0.125 | 0.06 | 2 | 0.25 | 0.25 | 1 |
| M76297 | 2000 | 0.06 | 0.03 | 2 | 0.25 | 0.125 | 2 |
| E46972 | >2000 | 1 | <0.002 | >512 | 0.25 | <0.008 | >32 |
| M36292 | 2000 | 0.06 | <0.002 | >32 | 0.25 | <0.008 | >32 |
| X7199 | >2000 | 16 | 16 | 1 | 0.25 | 0.125 | >32 |
| NCTC 11168 | 2000 | 0.06 | 0.03 | 2 | 0.25 | 0.06 | 4 |
| MIC$_{90}$[c] | 2000 | 16 | 4 | **4** | 256 | 16 | **16** |
| MIC$_{50}$[d] | 2000 | 0.25 | 0.06 | **4** | 0.5 | 0.06 | **8** |

[a] Addition of (subinhibitory) 125 mg/L (-)-α-pinene

[b] Fold change (improvement) of MIC with addition of 125 mg/L (-)-α-pinene

[c] Concentration that inhibits 90% of the tested strains [24]

[d] Concentration that inhibits 50% of the tested strains [24]

treatments with 62.5 mg/L and 125 mg/L (-)-α-pinene showed 83% (*p* = 0.001) and 85% (*p* <0.001) inhibition, respectively, of quorum sensing compared to the untreated control. These data thus showed concentration-dependent quorum-sensing inhibitory activities of (-)-α-pinene, which was emphasized by the prolonged treatment times.

## (-)-α-Pinene does not reduce fluoroquinolone resistance development when added to enrofloxacin in broiler chickens

Although the use of enrofloxacin is prohibited in poultry production in the USA due to its rapid induction of fluoroquinolone resistance in *C. jejuni* [25], it is still used in veterinary medicine in the European Union [26]. Based on the obvious activity of (-)-α-pinene in modulating fluoroquinolone resistance (Table 1), we investigated whether a subinhibitory concentration of (-)-α-pinene can delay development of fluoroquinolone resistance of *C. jejuni* in the chicken model, with the addition of (-)-α-pinene and enrofloxacin into the broilers water supply. Here, the (-)-α-pinene concentration used was doubled (but still subinhibitory) compared to the *in vitro* studies, to ensure sufficient concentration in the intestinal tract in chickens.

As treatments with enrofloxacin are known to induce fluoroquinolone resistance in *Campylobacter* [25], these analyses included enumeration of both the total numbers of *Campylobacter* and the proportion of ciprofloxacin-resistant *Campylobacter* in each broiler chicken fecal sample. The results showed that all of the *Campylobacter* isolates (100%), in all of the colonized birds treated with enrofloxacin developed fluoroquinolone resistance at all of the sampling times after the treatment had begun, regardless of inclusion of (-)-α-pinene in the drinking water. In contrast, *Campylobacter* isolates from the broiler chicken groups treated with DMSO and (-)-α-pinene alone (i.e., no enrofloxacin treatment) did not develop any resistance (0%) to

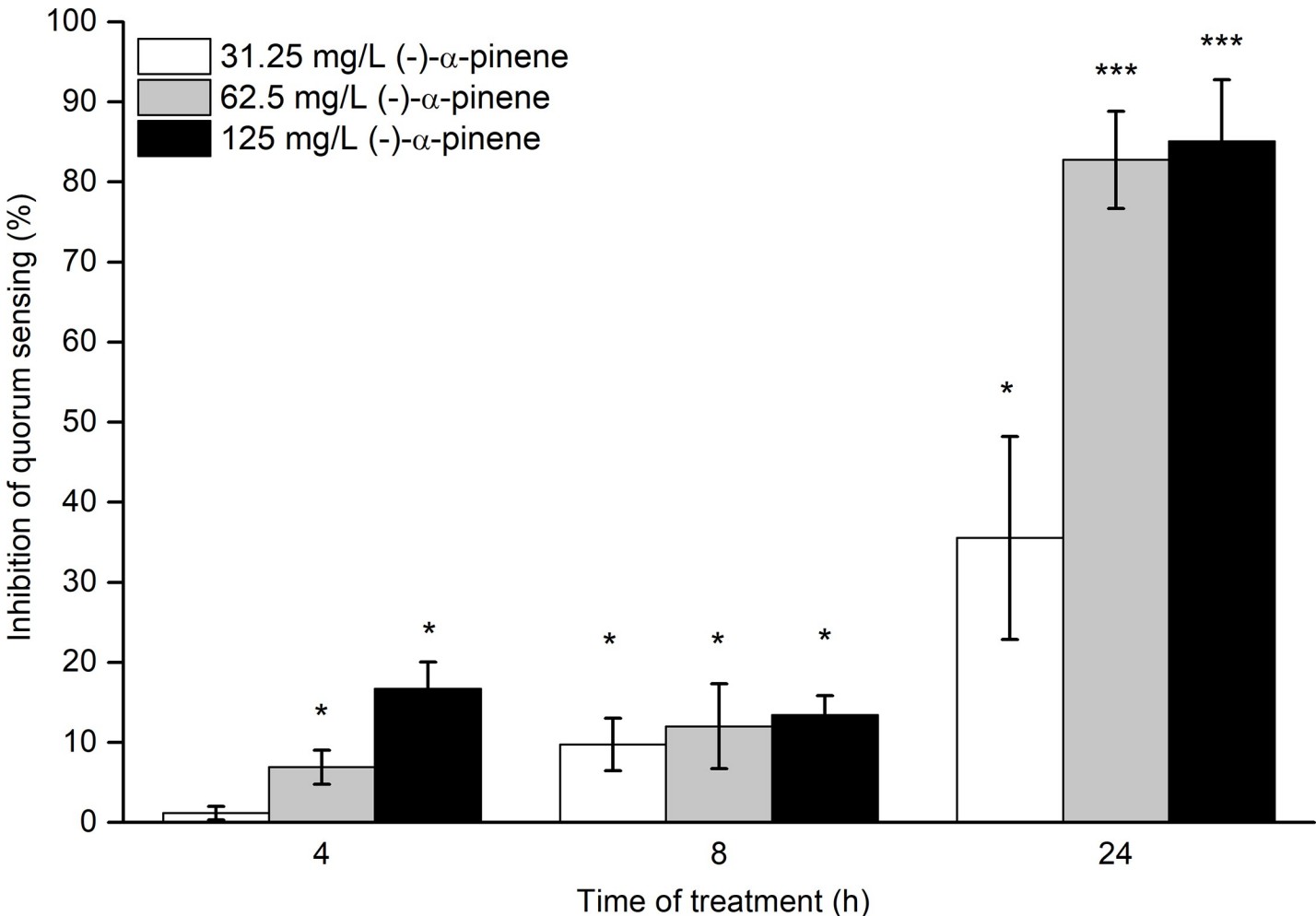

**Fig 1. Time-course of (-)-α-pinene inhibition of quorum sensing in *C. jejuni* NCTC11168.** Data are means ±standard deviation of relative reduction of quorum sensing signal (as luminescence of *V. harveyi* BB170) in the treated *C. jejuni* cell free supernatants (CFS) *versus* the untreated *C. jejuni* CFS, calculated from three replicates. * $p < 0.05$, *** $p \leq 0.001$.

fluoroquinolones at any point during the experiment. During these treatments, the mean *Campylobacter* numbers for (-)-α-pinene alone (4.37 $\log_{10}$ CFU/g) tended to be lower than that of the DMSO control (5.05 $\log_{10}$ CFU/g); however, there was wide variability within each of these treatment groups, so these data did not reach statistical significance (Fig 2).

These data showed that (-)-α-pinene did not have any resistance-modulatory activity *in vivo*, nor did it reduce the rapid development of fluoroquinolone resistance of *C. jejuni* after exposure to enrofloxacin. However, addition of (-)-α-pinene alone to the broiler chicken water supply reduced the average numbers of *Campylobacter* in the already colonized broiler chickens, although this did not reach statistical significance. Of note, with the (-)-α-pinene here added to the water supply, the amount of (-)-α-pinene ingested by each broiler chicken could not be controlled, which is likely to explain the large variations in these data.

### Reduction of *Campylobacter* in broiler chicken cecum after direct gavage with (-)-α-pinene

To better evaluate whether (-)-α-pinene can modulate *C. jejuni* colonization in the same broiler chicken experiment described above, chickens colonized by FQ-S *Campylobacter* and

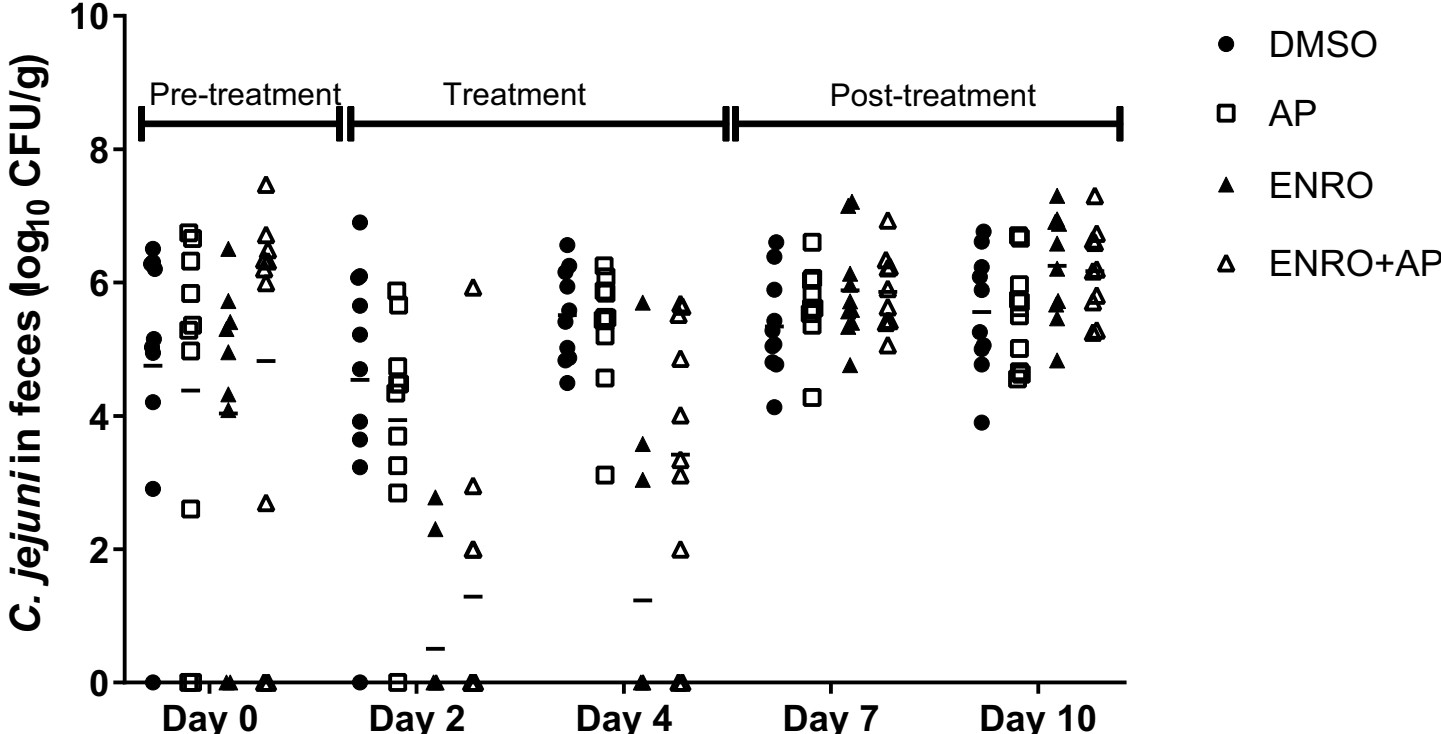

**Fig 2. Time-courses of the effects of (-)-α-pinene in the water supply of broiler chickens inoculated with *C. jejuni* NCTC 11168 3 days before (-)-α-pinene treatment (started day 0).** Data are *C. jejuni* counts (log$_{10}$ CFU/g feces) in cloacal swabs from individual broiler chickens in the treatment groups: DMSO, no treatment control; AP, 250 mg/L (-)-α-pinene; ENRO, 50 mg/L enrofloxacin; ENRO+AP, combination of 50 mg/L enrofloxacin and 250 mg/L (-)-α-pinene. The detection limit of the culture method was approximately 2 log$_{10}$ CFU/g feces, and the means are indicated by the horizontal lines.

FQ-R *Campylobacter* were treated with (-)-α-pinene by direct oral gavage. With this treatment, the amount of (-)-α-pinene consumed by each animal was better controlled.

These data indicated that there were significantly lower *C. jejuni* counts in the broiler chickens colonized with FQ-S *C. jejuni* when treated with (-)-α-pinene (FQ-S treated), with a reduction of 0.8 log$_{10}$ CFU/g unit ($p = 0.028$; Fig 3) compared to the untreated group (FQ-S untreated). No significant differences were seen between the nontreated and (-)-α-pinene–treated groups that were colonized with FQ-R *C. jejuni* (FQ-R untreated, FQ-R treated), although a slight mean reduction from 7.0 to 6.6 log$_{10}$ CFU/g cecum content ($p = 0.095$; Fig 3) was observed in the treated group compared to untreated. These data show that (-)-α-pinene reduced colonization of the FQ-S *C. jejuni* in broiler chickens when administered by direct gavage, but it had no significant effect on FQ-R *C. jejuni*.

## Discussion

The effects of (-)-α-pinene have been seen to be versatile, from antioxidative to cell protective [27], and to anti-cancer [28], with only weak antimicrobial activities reported previously [29]. In the present study, we showed that the concentrations of (-)-α-pinene needed for antimicrobial effects against *C. jejuni* were high, and were similar for all of the strains (n = 57) tested regardless of host origin (e.g., chicken, turkey, human) and susceptibility profiles to a range of antibiotics. These data confirm the observations of Kovač et al. [21], where (-)-α-pinene also showed weak antimicrobial activity against nine *C. jejuni* strains tested. We also confirmed earlier indications of the strong *in-vitro* resistance-modulatory activity of (-)-α-pinene for

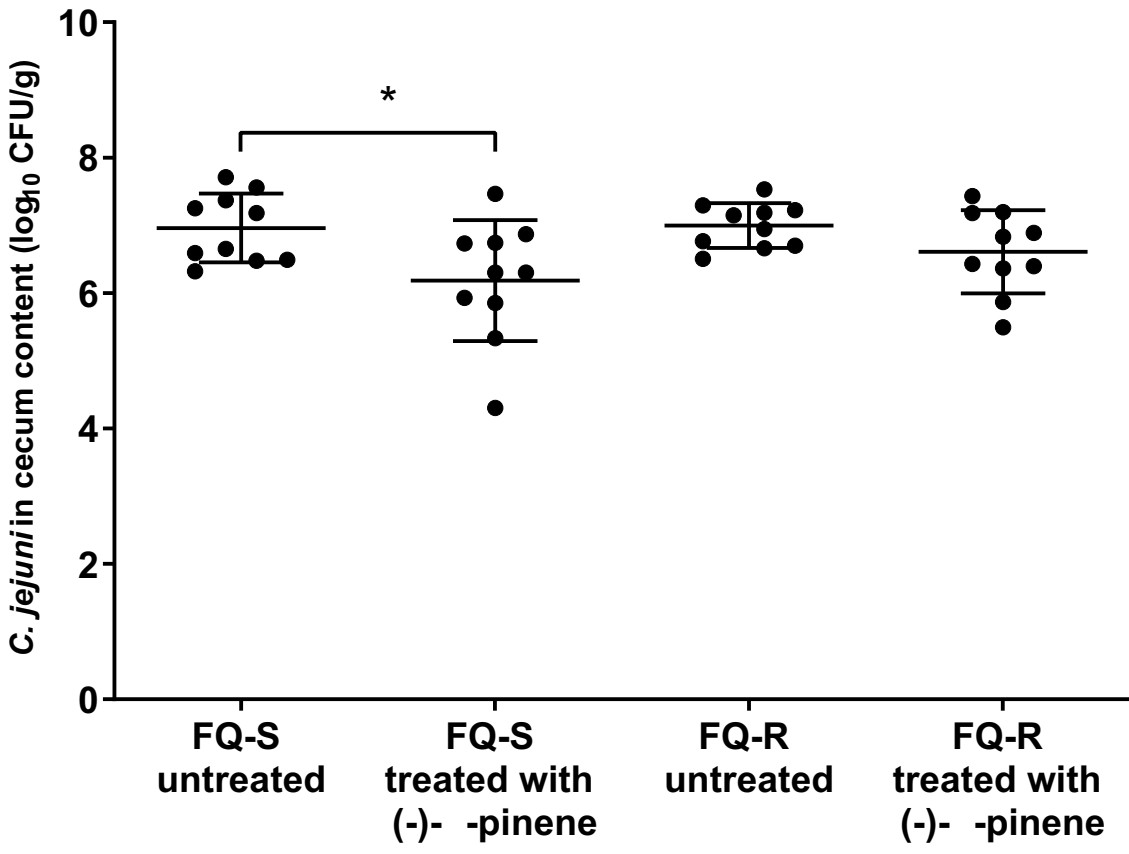

**Fig 3. *Campylobacter* counts ($\log_{10}$ CFU/g) in the cecum content of broiler chickens following treatments without and with a subinhibitory (-)-α-pinene concentration via direct gavage for 3 consecutive days (0.4 mL, 250 mg/L, daily).** The samples were collected 24 h after the last treatment, and cultured for *Campylobacter*. Data are means ±standard deviation from individual broiler chickens that were colonized with either FQ-S or FQ-R *C. jejuni* prior to the treatment. FQ-S/FQ-R untreated, fluoroquinolone-sensitive/resistant *C. jejuni* (controls); FQ-S/FQ-R treated, fluoroquinolone-sensitive/resistant *C. jejuni* plus treatment with (-)-α-pinene. *$p < 0.05$.

clinically important antibiotics (i.e., ciprofloxacin and erythromycin) against this large and diverse collection of *C. jejuni* strains [21]. These data suggest that (-)-α-pinene may have potential as an adjunctive therapy, in various hosts, to increase the efficacy of macrolides and fluoroquinolones against *C. jejuni* resistant to these antimicrobials.

Subinhibitory concentrations of (-)-α-pinene have been shown to evoke diverse transcriptional responses in *C. jejuni*, although the main mechanisms of its resistance-modulatory activity appear to be inhibition of the CmeABC efflux pump and induction of membrane damage [21]. Similarly, Oh and Jeon [30] reported that different monoterpenes can show synergistic effects when combined with ciprofloxacin or erythromycin, due to the modulation of antibiotic influx and efflux in *C. jejuni*.

Correct functioning of the efflux pumps, such as the CmeABC multidrug efflux pump (the major efflux pump in *C. jejuni*), is needed not only to enhance bacterial resistance to antibiotics, but to also increase bacterial resistance to bile salts, and thus to facilitate the colonization of the gastrointestinal tract in animals and humans by *C. jejuni* [31]. This suggests that after exposure to an efflux pump inhibitor (e.g., pinene), *C. jejuni* sensitivity to antimicrobials can increase, its virulence can decrease, and its colonization can become impaired [32]. We therefore tested the potential of (-)-α-pinene to modulate *C. jejuni* resistance to fluoroquinolones,

to attenuate the development of resistance to fluoroquinolones and impair *C. jejuni* colonization *in vivo* in a broiler chicken model.

In the broiler chicken model, (-)-α-pinene did not act as a modulator of *C. jejuni* resistance when administered together with enrofloxacin, nor did it change the development of resistance in *C. jejuni* to fluoroquinolones when NCTC 11168 was used as the model strain (Fig 2). Due to the ever-growing antibiotic resistance of *C. jejuni* [33], the concept of a natural compound that can hinder resistance development or have a synergistic activity with antibiotics would open new and attractive opportunities for combating antibiotic resistance. However, the reality is often more complex, as compounds that demonstrate activity *in vitro* do not always maintain the same activity *in vivo*, as additional factors, that cannot be controlled, are introduced.

It has been suggested that a reduction of *Campylobacter* in the chicken intestine by 1 $\log_{10}$ CFU can reduce the public health risk by 50% to 90%, and a 2 $\log_{10}$ CFU reduction can reduce the risk by >90% [8,34,35]. This can be achieved with natural compounds [10]. Supplementation of poultry feed and water with natural compounds has been shown to reduce *Campylobacters* in poultry, and in some cases, this has improved animal health and yield as well. For example, feed supplementation with carvacrol and thymol at inhibitory concentrations has shown significant reduction of *Campylobacter* and *Salmonella* colonization [36] and growth enhancement in broiler chickens [37]. Grilli et al. [9] lowered *Campylobacter* counts in the broiler chicken cecum by 1 $\log_{10}$ CFU/g with feed additives of essential oils at 5000 mg/L, which represented an antimicrobial concentration. In the present study, a 0.8 $\log_{10}$ CFU reduction in *Campylobacter* counts in the broiler chicken cecum was obtained using a lower, and subinhibitory, concentration of (-)-α-pinene (250 mg/L) (Fig 3). This suggests that even lower concentrations of natural compounds, where bioactivity is still observed, can contribute to *Campylobacter* control in broilers, thus reducing the amount of treatment needed. The *C. jejuni* colonization reduction by (-)-α-pinene in a subinhibitory concentration can be explained by its efflux pump inhibitory [21] and quorum sensing inhibitory (Fig 1) activities exhibited at sub-inhibitory concentrations *in vitro*. In *C. jejuni*, both intact efflux pump activity [31] and quorum sensing [17] are important for colonization of the host, thus can the inhibition of these systems, by an external source such as (-)-α-pinene, contribute to *C. jejuni* control.

When a substance is introduced into an animal host to promote the reduction of pathogens, it is important to consider both the host response [38] and the response of the pathogen in question to the substance. An important factor for *C. jejuni* host colonization is cell-to-cell communication, or quorum sensing [13,39]. Disruption of the quorum-sensing system of *C. jejuni* interferes with its motility and autoagglutination, its production of cytolethal distending toxin, and its host colonization [17,18].

Essential oils and their constituents, such as cinnamaldehyde and cinnamon bark essential oil, can inhibit quorum sensing [40]. Furthermore, in *C. jejuni*, quorum-sensing inhibitors such as epigallocatechin gallate and extracts of *Euodia ruticarpa* can reduce motility and biofilm formation [19,20], although the potential *in-vivo* effects of these on *C. jejuni* colonization are not known. Brackman et al. [40] demonstrated 65% inhibition of quorum sensing by cinnamaldehyde in *Vibrio* spp. This was lower compared to that of (-)-α-pinene against *C. jejuni* in the present study, where the inhibition was >80% when treated with subinhibitory (-)-α-pinene (Fig 1). The quorum-sensing inhibition in *Vibrio* spp. resulted in down-regulation of virulence factors and weaker cytotoxicity toward *Caenorhabditis elegans* [40], which indicated that cinnamaldehyde can modulate the pathogen–host interactions. In the present study, we observed changes in pathogen–host interactions in terms of *C. jejuni* colonization in broiler chicken cecum content after treatment with (-)-α-pinene by direct gavage (Fig 3).

Although the anti-*Campylobacter* activity of (-)-α-pinene under *in vitro* conditions was similar against both ciprofloxacin-susceptible and -resistant *C. jejuni* strains (Table 1), the reduction of FQ-R *Campylobacters* in the broiler chickens treated with (-)-α-pinene did not reach significance. These findings can be explained by the observations of Luo et al. [25], who showed that FQ-R *C. jejuni* strains have better fitness *in vivo* compared to FQ-S strains, and are thus more tolerant to stressors, such as (-)-α-pinene treatment.

The present study stresses the importance of improving the control of *Campylobacter* in poultry production so as to reduce the public health risk. It also exposes the problem of development of antibiotic resistance in poultry production and the difficulties in the management of human foodborne infections by antibiotic-resistant *C. jejuni*. Further investigations into the mechanisms of action of natural compounds that might be used for manipulation of pathogen–host interactions and reduction of host colonization *in vivo* is highly warranted. Furthermore, it is important to consider the effects and mechanisms of action of such compounds at subinhibitory concentrations. This can allow improved prediction of their activity in live systems (i.e., in the animal), where it can be difficult to control the exact amounts that are ingested. For example, for *Pseudomonas aeruginosa*, inhibitors of quorum sensing have been shown to mitigate infections even without showing strong antibacterial effects [41,42]. This makes the inhibitory effects on quorum sensing an important aspect when searching for new anti-infectious compounds.

For (-)-α-pinene, in *C. jejuni* it was shown previously to evoke stress and heat-shock responses, to inhibit the multidrug efflux pumps, and to increase membrane permeability [21]. The present study further indicates that it can inhibit quorum sensing in *C. jejuni*. It is likely that all of these *in vitro* activities of (-)-α-pinene might have contributed to its *in vivo* effects observed in the current study on colonization by fluoroquinolone-susceptible *C. jejuni* in chickens.

## Conclusions

The findings from this study indicate that despite showing poor antimicrobial activity against *Campylobacter*, even at high concentrations, (-)-α-pinene can modulate *C. jejuni* quorum sensing and colonization of broiler hosts when administered at subinhibitory concentrations. Further *in vivo* studies are warranted to better evaluate the effects of (-)-α-pinene on colonization by *Campylobacter*, including different species and strains with different antimicrobial resistance profiles (e.g., erythromycin resistance), as well as various treatment regimens (e.g., therapeutic vs. preventive).

## Supporting information

**S1 Fig. Luminescence production (relative luminescent units, RLU; open squares) and growth of *V. harveyi* BB170 (OD$_{600nm}$; filled circles) in AB medium supplemented with 20% MHB.** Data are means ±standard deviation. Framed time points (i.e. 9 h) are those best suited for further evaluation of quorum sensing inhibition.
(DOCX)

**S2 Fig. Luminescence of *V. harveyi* BB170 produced after addition of cell-free supernatants of *C. jejuni* wild type (wt control; filled symbols) and the Δ*luxS* mutant (*luxS*; open symbols) without treatment (circles) and after treatment with (-)-α-pinene at 31.25 mg/L (squares), 62.5 mg/L (triangles), and 125 mg/L (diamonds).** Data are means ±standard deviation for luminescent signal (relative luminescent units, RLU). Framed time point is that used

in the calculation of quorum sensing inhibition.
(DOCX)

**S1 Table. *Campylobacter jejuni* strains used in this study.**
(DOCX)

**S2 Table. Susceptibility of *Campylobacter jejuni* broiler and turkey strains to the range of tested antibiotics, as MICs and corresponding sensitivity (S) or resistance (R).**
(DOCX)

## Author Contributions

**Conceptualization:** Katarina Šimunović, Jasna Kovač, Qijing Zhang.

**Formal analysis:** Katarina Šimunović, Zhangqi Shen.

**Funding acquisition:** Anja Klančnik, Qijing Zhang, Sonja Smole Možina.

**Investigation:** Orhan Sahin.

**Methodology:** Katarina Šimunović, Orhan Sahin.

**Resources:** Qijing Zhang, Sonja Smole Možina.

**Supervision:** Orhan Sahin, Zhangqi Shen, Sonja Smole Možina.

**Validation:** Katarina Šimunović, Zhangqi Shen.

**Visualization:** Katarina Šimunović.

**Writing – original draft:** Katarina Šimunović, Orhan Sahin.

**Writing – review & editing:** Katarina Šimunović, Orhan Sahin, Jasna Kovač, Zhangqi Shen, Anja Klančnik, Qijing Zhang, Sonja Smole Možina.

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
