## [Decision Letter · Decision Letter 0]

16 Oct 2019

PONE-D-19-20048

(-)-α-Pinene reduces quorum sensing and Campylobacter jejuni colonization in broiler chickens

PLOS ONE

Dear Dr. Smole Možina,

Thank you for submitting your manuscript to PLOS ONE. After careful consideration, we feel that it has merit but does not fully meet PLOS ONE’s publication criteria as it currently stands. Therefore, we invite you to submit a revised version of the manuscript that addresses the points raised during the review process.

As is often the case, the two reviewers have highlighted different points in their reviews.  I would urge the authors to read these comments carefully for any revision they make, as there are quite a few of them.  However, in terms of what needs to be addressed, the following four points can be made:

1)    Methodological issues - sampling as swabs, but reporting data as if it were faeces, and concerns over the impact of outlier data in the conclusions drawn

2)    Repetition of methods and results - both reviewers comment on this

3)    Justification for issues around the study design - relevance of the study to humans in terms of drug use in humans and chickens

4)    Discussion - some clarification over aspects in regards to framing the results in the current literature

We would appreciate receiving your revised manuscript by Nov 30 2019 11:59PM. To enhance the reproducibility of your results, we recommend that if applicable you deposit your laboratory protocols in protocols.io, where a protocol can be assigned its own identifier (DOI) such that it can be cited independently in the future. For instructions see: http://journals.plos.org/plosone/s/submission-guidelines#loc-laboratory-protocols

We look forward to receiving your revised manuscript.

Kind regards,

Patrick Jon Biggs, PhD

Academic Editor

PLOS ONE

Journal Requirements:

2. In your Methods section, please provide additional details regarding the animals used in your study and ensure you have described the source. For more information regarding PLOS' policy on materials sharing and reporting, see https://journals.plos.org/plosone/s/materials-and-software-sharing#loc-sharing-materials.

Reviewers' comments:

Reviewer's Responses to Questions

**Comments to the Author**

1. Is the manuscript technically sound, and do the data support the conclusions?

Reviewer #1: Partly

Reviewer #2: Yes

2. Has the statistical analysis been performed appropriately and rigorously? 

Reviewer #1: No

Reviewer #2: Yes

3. Have the authors made all data underlying the findings in their manuscript fully available?

Reviewer #1: Yes

Reviewer #2: Yes

4. Is the manuscript presented in an intelligible fashion and written in standard English?

Reviewer #1: No

Reviewer #2: Yes

5. Review Comments to the Author

Reviewer #1: This paper presents valuable and to a large extent confirmatory findings on the effect of the essential oil component (-)-α-pinene against the most important foodborne pathogen Campylobacter jejuni. The effect of this component on the pathogen was demonstrated on different levels: in vitro through antimicrobial activity, antibiotic resistance modulation, and quorum sensing inhibition; in vivo through a chicken infection model.

The paper is largely based on a previous study of the same research groups (Kovac et al. 2015) in which both the antimicrobial activity and the resistance modulatory activities of (-)-α-pinene in C. jejuni against two fluoroquinolones was investigated. In the present paper, the C. jejuni strain collection was extended to confirm these activities, but surprisingly the strain (NCTC 11168) used for inoculation of the chickens was not included. In addition, now also the inhibition of quorum sensing was shown in a Vibrio harveyi model. Based on these observations, a broiler chicken infection model was used to see if (-)-α-pinene added in the drinking water would reduce the C. jejuni load in the gut. In contract to the in vitro results, no resistance modulatory effect was seen in the chickens, but where are the in vitro data which proof this effect for the inoculation strain ? No effect, either (-)-α-pinene alone or in combination with enrofloxacin, was seen on the colonization level. Only through a direct gavage of (-)-α-pinene, a statistically significant but very modest reduction (< 1 log cfu/g feces) was observed for the ciprofloxacin sensitive C. jejuni population only. In fact, for half of the chickens infected with this sensitive C. jejuni population, the C. jejuni excretion level was comparable with the untreated control. The statistical significance was probably only obtained through one bird showing a considerable lower C. jejuni level. I therefore doubt about the soundness of the statistical analysis using a t-test and the inclusion of the outlier bird.

Although the different categories of results are valuable on their own, there is a general lack of coherence between the results. The quorum sensing results have almost no connection with the other types of data and as they are only observed in vitro, there is no indication that this activity will also appear in vivo and therefore would help to find the mechanism of the observed C. jejuni colonization decrease with (-)-α-pinene direct gavage.

The rationale to look for a synergy between (-)-α-pinene and enrofloxacin in broiler chickens is not clear. Enrofloxacin is used in human medicine, but is not used in broilers to prevent or combat a C. jejuni infection. The observation of the enrofloxacin/ciprofloxacin resistance modulation by (-)-α-pinene in vitro is interesting and could have practical considerations for human clinical use, but I doubt about the practical relevance in poultry rearing.

The introduction is written too general about the control of C. jejuni in poultry and the increasing risk of AMR in this pathogen. As a result, the real focus of this paper which is the effect of (-)-α-pinene on C. jejuni is not well introduced. For example, quorum sensing which is one of the focus points, is only introduced with a small paragraph of 5 lines, but mentioning only one reference related to C. jejuni in a guinea pig abortion model which is not really relevant for poultry.

The discussion is written too much as a review of literature on the effect of other essential oils or plant components on pathogens and on the importance of improving the control of C. jejuni in poultry in general. It’s not enough focused on the explanation and discussion of the own results. For example, there is no hypothesis why (-)-α-pinene has no effect on the ciprofloxacin resistant C. jejuni population in the broiler infection model.

There are still some corrections in English to be made, this should be checked throughout the manuscript.

Other specific remarks:

-L145: for what purpose was the NCTC 11168 luxS mutant used? As negative control?

-L156: explain AB

-L160: is the negative control correct here (V. harveyi suspension)?

-L166 & 172: incorrect numbering of figures (Fig. S2 before S1)

-L175-176: there can’t be two negative controls, the 11168 luxS mutant and fresh MHB?

-L191: explain MHA-SS; details for cloacal swabs. Campylobacter status of chickens was only checked by direct plating, not by enrichment? What was the detection limit?

-L218: plated and incubated

-L258, table 1: host origin of the strains can be shown here

-L275: strains vs. isolates

-L280: also strain W14861 has this behavior

-L311-312: I don’t understand how the results are presented here versus the negative control (luxS mutant)

-L324-328: this is a repetition from M&M

-L341: where are the horizontal lines in the figure?

-L346: in Fig. 2, I only see 3 colonized birds

-L349: idem, only 7 colonized birds?

-L366: what is the exact meaning of “the potential to lower…”, this is scientifically not well formulated; is it statistically significant or only a trend?

-L373-385: this is a repetition from M&M

-L439: macrolides were not included here

Reviewer #2: The subject of the work is interesting and innovative. It raises an important issue regarding the search for alternative methods to combat Campylobacter in poultry. It would also be worth checking (maby in next stage of research) what effect the (-)-a-pinene has on other microorganisms inhabiting the intestines in birds. Does it not inhibit the growth and colonization of beneficial microflora such as lactobacilli or bifidocabterie?

I have some reservations about the methodology of the experiment regarding the assessment of the influence of a-pinene on the colonization of Camylobacter in chicks. The authors took cloacal swabs, diluted in MHB and plated on agar; the result is given in cfu/g feces. How did the authors weigh droppings when they took a swab?

The authors also tend to repeat in the Results chapter information from the Materials and Methods chapter, e.g. P17L324-328; P19L373-377.

6. PLOS authors have the option to publish the peer review history of their article (what does this mean?). If published, this will include your full peer review and any attached files.

Reviewer #1: No

Reviewer #2: No

---

## [Author Response · Author response to Decision Letter 0]

30 Nov 2019

Response to reviewers

Title of manuscript: (-)-α-Pinene reduces quorum sensing and Campylobacter jejuni colonization in broiler chickens 

Reviewer #1

1. The paper is largely based on a previous study of the same research groups (Kovac et al. 2015) in which both the antimicrobial activity and the resistance modulatory activities of (-)-α-pinene in C. jejuni against two fluoroquinolones was investigated. In the present paper, the C. jejuni strain collection was extended to confirm these activities, but surprisingly the strain (NCTC 11168) used for inoculation of the chickens was not included.

Answer: The strain used for the chicken inoculation was also used in the in-vitro resistance modulatory part of the study. In Table 1, the last strain under the designation “11168” is the inoculation strain. We have changed this designation to “NCTC 11168” now, to avoid further confusion.

2. In addition, now also the inhibition of quorum sensing was shown in a Vibrio harveyi model. Based on these observations, a broiler chicken infection model was used to see if (-)-α-pinene added in the drinking water would reduce the C. jejuni load in the gut. In contrast to the in vitro results, no resistance modulatory effect was seen in the chickens, but where are the in vitro data which proof this effect for the inoculation strain ?

Answer: Please note that in Table 1, where the in-vitro resistance modulatory data are shown, the data for the inoculation strain (NCTC 11168) are shown in the bottom part. The resistance against ciprofloxacin showed a 2-fold change, and the resistance against erythromycin showed a 4-fold change.

3. No effect, either (-)-α-pinene alone or in combination with enrofloxacin, was seen on the colonization level. Only through a direct gavage of (-)-α-pinene, a statistically significant but very modest reduction (< 1 log cfu/g feces) was observed for the ciprofloxacin sensitive C. jejuni population only. In fact, for half of the chickens infected with this sensitive C. jejuni population, the C. jejuni excretion level was comparable with the untreated control. The statistical significance was probably only obtained through one bird showing a considerable lower C. jejuni level. I therefore doubt about the soundness of the statistical analysis using a t-test and the inclusion of the outlier bird.

Answer: As the data has a normal distribution, which was tested with Kolmogorov-Smirnov and Shapiro-Wilk tests (see Table1), students t-test is the appropriate statistical test here. We have now compared the statistical significance of the data with all of the data pointes (Table 2) and without the outlier (Table3, and the statistical significance, albeit weaker, still remains.

Table 1. Normality tests.

Tests of Normality

 Group Kolmogorov-Smirnova Shapiro-Wilk

 Statistic df Sig. Statistic df Sig.

CjejuniNo cipS 0.228 10 0.151 0.900 10 0.221

 cipS-AP 0.156 10 0.200 0.947 10 0.628

 cipR 0.179 10 0.200 0.954 10 0.715

 cipR-AP 0.145 10 0.200 0.952 10 0.687

Table 2. T-tests with all of the data.

 Levene's Test for Equality of Variances t-test for Equality of Means

CjejuniNo F Sig. t df Sig. (2-tailed) Mean Difference Std. Error Difference 95% Confidence Interval of the Difference

 Lower Upper

Equal variances assumed 1.287 0.271 2.397 18 0.028 0.7788 0.325 0.09613 1.462

Equal variances not assumed 2.397 14.26 0.031 0.7788 0.325 0.08307 1.475

Table 3. T-tests without outlier in the FQ-S (CipS) -AP group.

 Levene's Test for Equality of Variances t-test for Equality of Means

Nooutlier F Sig. t df Sig. (2-tailed) Mean Difference Std. Error Difference 95% Confidence Interval of the Difference

 Lower Upper

Equal variances assumed 0.121 0.732 2.166 17 0.045 0.56984 0.26306 0.01483 1.12485

Equal variances not assumed 2.139 15.302 0.049 0.56984 0.26637 0.00305 1.13662

4. Although the different categories of results are valuable on their own, there is a general lack of coherence between the results. The quorum sensing results have almost no connection with the other types of data and as they are only observed in vitro, there is no indication that this activity will also appear in vivo and therefore would help to find the mechanism of the observed C. jejuni colonization decrease with (-)-α-pinene direct gavage. “

Answer: The Introduction (L62-L63, L81-L88) has been changed and hopefully these changes will clarify and link the quorum sensing better with the rest of the data. We have explored quorum sensing inhibition as one of the possible mechanisms of (-)-α-pinene activity, together with the (previously published) efflux pump inhibition and membrane disruption. We are not proposing that quorum sensing inhibition is solely responsible for C. jejuni reduction after treatment with (-)-α-pinene, but rather that it is one of the actions of (-)-α-pinene against C. jejuni (L436 –L444).

5. The rationale to look for a synergy between (-)-α-pinene and enrofloxacin in broiler chickens is not clear. Enrofloxacin is used in human medicine, but is not used in broilers to prevent or combat a C. jejuni infection. The observation of the enrofloxacin/ciprofloxacin resistance modulation by (-)-α-pinene in vitro is interesting and could have practical considerations for human clinical use, but I doubt about the practical relevance in poultry rearing. 

Answer: (-)-α-pinene was shown to modulate C. jejuni resistance to fluoroquinolone antibiotics (Table 1) and was expected to affect the emergence of fluoroquinolone resistant Campylobacter during antibiotic treatment. Thus a rationale to examine a synergy between (-)-α-pinene and enrofloxacin was warranted. We have modified the manuscript to clarify this point. The chicken model is a well established in vivo system to study fluoroquinolone resistance development in Campylobacter, which also occurs in human patients treated with a fluoroquinolone antibiotic. Thus the finding is relevant to human clinical cases where fluoroquinolones are commonly used to treat campylobacteriosis. Additionally, enrofloxacin is still being used in poultry (and other animals) in certain parts of the world for treatment of bacterial infections, such as colibasillosis and Mycoplasma infections. Due to prevalence of C. jejuni in poultry intestinal tract as a commensal, the treatment of poultry with enrofloxacin will result in resistance development in C. jejuni, which can be subsequently transmitted to humans via the food chain. If (-)-α-pinene is found to be effective in inhibiting resistance development in Campylobacter, it may be used as adjunct therapy to control the development of antibiotic-resistant Campylobacter in different hosts. In this study we used only one strain for the in vivo experiment. Additional work needs to be done to examine the potential of (-)-α-pinene.

6. The introduction is written too general about the control of C. jejuni in poultry and the increasing risk of AMR in this pathogen. As a result, the real focus of this paper which is the effect of (-)-α-pinene on C. jejuni is not well introduced. For example, quorum sensing which is one of the focus points, is only introduced with a small paragraph of 5 lines, but mentioning only one reference related to C. jejuni in a guinea pig abortion model which is not really relevant for poultry. 

Answer: Thank you for this observation. We have edited the Introduction (L81-L98) and hope that this clarifies better the background and purpose of this study.

7. The discussion is written too much as a review of literature on the effect of other essential oils or plant components on pathogens and on the importance of improving the control of C. jejuni in poultry in general. It’s not enough focused on the explanation and discussion of the own results. For example, there is no hypothesis why (-)-α-pinene has no effect on the ciprofloxacin resistant C. jejuni population in the broiler infection model. 

Answer: We have changed some parts of the Discussion for clarification (L381-L383; 431-432: 439-440), although we feel that the comparison of pinene with essential oils and their constituents is necessary as pinene is the major constituent of many essential oils.

We have attempted to explain the lack of effect of pinene on FQ-R strains by the increased fitness of FQ-R strains observed by Luo et al. (2005) (L430-L436). We feel that more speculation about the reason for the weaker effect observed here would be unwise, as we do not have more information at this point. 

8. There are still some corrections in English to be made, this should be checked throughout the manuscript. 

Answer: The manuscript has been proofread and edited by a professional mother-tongue scientific editor.

9. Other specific remarks:

-L145: for what purpose was the NCTC 11168 luxS mutant used? As negative control?

Answer: Yes, it was used as a negative control. This has now been corrected in the manuscript.

-L156: explain AB

Answer: AB medium is described in the Bacterial strains and growth conditions (L111-L114).

-L160: is the negative control correct here (V. harveyi suspension)?

Answer: No, this was an error, and we apologize for it. It has now been removed from the manuscript.

-L166 & 172: incorrect numbering of figures (Fig. S2 before S1)

Answer: Thank you for this observation. The labeling has been corrected in text, and the Figure labels have remained the same.

-L175-176: there can’t be two negative controls, the 11168 luxS mutant and fresh MHB?

Answer: This is correct, and the error has been corrected. MHB is the “blank” value.

-L191: explain MHA-SS?

Answer: MHA-SS is described in Bacterial strains and growth conditions: L113-L115.

-details for cloacal swabs

Answer: We considered approximately 100 mg of feces per cloacal swab, and we have presented the data accordingly. 

This is the method previously used in other studies (e.g., Sahin et al. (2003) doi: 10.1128/AEM.69.9.5372-5379.2003, and Luo et al. (2003) doi:10.1128/AAC.47.1.390-394.2003), and to maintain consistency and for better comparisons of our data with already published data, we would like to keep this as it is. The description has been added into the manuscript for clarification (L215-L220). Cloacal swabs are commonly used to detect Campylobacter shedding in poultry in a non-invasive way, as was used in the present study.

-Campylobacter status of chickens was only checked by direct plating, not by enrichment? What was the detection limit

Answer: Direct plating is considered a better option for Campylobacter spp. isolation from chicken feces, compared to enrichment (Musgrove et al., 2001; Sahin et al., 2003). The detection limit was 100 CFU/g feces. The description of the method has been adjusted for better clarification (L215-L220).

Musgrove, M.T., M.E. Berrang, J.A. Byrd, N.J. Stern, and N.A. Cox. 2001. Detection of Campylobacter spp. in ceca and crops with and without enrichment. Poult Sci. 80:825–828.

Sahin, O., Q. Zhang, and T.Y. Morishita. 2003. Detection of Campylobacter. In: Microbial Food Safety in Animal Agriculture. M.E. Torrence and R.E. Isaacson, eds. Iowa State Press, Ames, IA.183–193

-L218: plated and incubated

Answer: The text has been added to L218-L219.

-L258, table 1: host origin of the strains can be shown here

Answer: The description has been added as suggested.

-L275: strains vs. isolates

Answer: Corrected.

-L280: also strain W14861 has this behavior

Answer: Thank you for the observation. This has been corrected (L285).

-L311-312: I don’t understand how the results are presented here versus the negative control (luxS mutant)

Answer: The error was corrected for L314-317.

-L324-328: this is a repetition from M&M

Answer: Corrected.

-L341: where are the horizontal lines in the figure?

Answer: In each stack of data points there is a (small) horizontal line that represents the means of each dataset. Due to some technical difficulties with the Figure format, this was not presented properly. This has now been corrected.

-L346: in Fig. 2, I only see 3 colonized birds

Answer: Four birds are colonized, but the symbols are overlapping. The symbol with the thicker edges (the lowest) is actually two symbols. This could not be corrected on this graph.

-L349: idem, only 7 colonized birds?

Answer: As in the previous point, here we have two symbols on the upper part that are overlapping, and thus look like one symbol with thicker edges.

-L366: what is the exact meaning of “the potential to lower…”, this is scientifically not well formulated; is it statistically significant or only a trend?

Answer: This has been corrected.

-L373-385: this is a repetition from M&M

Answer: Corrected. 

-L439: macrolides were not included here 

Answer: Corrected. The sentence has been rearranged for clarification.

Reviewer #2

1. The subject of the work is interesting and innovative. It raises an important issue regarding the search for alternative methods to combat Campylobacter in poultry. It would also be worth checking (maby in next stage of research) what effect the (-)-a-pinene has on other microorganisms inhabiting the intestines in birds. Does it not inhibit the growth and colonization of beneficial microflora such as lactobacilli or bifidocabterie?

Answer: Thank you for this observation. Any treatment added to chicken water that can affect C. jejuni numbers will probably also influence other members of the intestinal microbiota, although it does not necessarily mean a negative influence on positive microbes, as are bifidobacteria. This is something that will indeed be considered for future work.

2. I have some reservations about the methodology of the experiment regarding the assessment of the influence of a-pinene on the colonization of Camylobacter in chicks. The authors took cloacal swabs, diluted in MHB and plated on agar; the result is given in cfu/g feces. How did the authors weigh droppings when they took a swab?

Answer: Each swab diluted was considered to have approximately 100 mg of fecal matter and was diluted in 1 mL of MHB so as to get a 10-fold dilution. Dilutions were plated and data are shown as cfu/g feces, with a detection limit of 100 cfu/g feces. We considered 100 mg of feces per cloacal swab and presented the data accordingly. 

This is the method previously used in other studies (e.g., Sahin et al. (2003) doi:10.1128/AEM.69.9.5372-5379.2003 and Luo et al. (2003) doi:10.1128/AAC.47.1.390-394.2003, and for consistency and better comparisons of our data with already published data we prefer to keep this as it is. A description has been added into the manuscript for clarification (L209-L212). Cloacal swabs are commonly used to detect Campylobacter shedding in poultry in a non-invasive way, and it was used in the present study as such.

3. The authors also tend to repeat in the Results chapter information from the Materials and Methods chapter, e.g. P17L324-328; P19L373-377.

Answer: Corrections have been made to these parts of the text.

Editor

As is often the case, the two reviewers have highlighted different points in their reviews. I would urge the authors to read these comments carefully for any revision they make, as there are quite a few of them. However, in terms of what needs to be addressed, the following four points can be made:

1) Methodological issues - sampling as swabs, but reporting data as if it were faeces, and concerns over the impact of outlier data in the conclusions drawn

Answer: Corrections and additional explanations have been added into the text to explain the swabbing process better. 

Additional statistical analysis was performed using all of the data and the data without the outlier, and the statistical significance remained. So we believe that the effect of pinene shows real significance.

2) Repetition of methods and results - both reviewers comment on this

Answer: This has been corrected.

3) Justification for issues around the study design - relevance of the study to humans in terms of drug use in humans and chickens

Answer: Corrections have been made to the text to clarify this matter. We do not propose that enrofloxacin should be used for Campylobacter control in poultry, and we apologize for the confusion. 

4) Discussion - some clarification over aspects in regards to framing the results in the current literature

Answer: Some modifications have been made throughout the Discussion (L395-L398; 435-443: 486-488).

Journal requirements

Please ensure that your manuscript meets PLOS ONE's style requirements, including those for file naming. The PLOS ONE style templates can be found at http://www.journals.plos.org/plosone/s/file?id=wjVg/PLOSOne_formatting_sample_main_body.pdf and http://www.journals.plos.org/plosone/s/file?id=ba62/PLOSOne_formatting_sample_title_authors_affiliations.pdf

Answer: Corrections have been made throughout the manuscript. 

2. In your Methods section, please provide additional details regarding the animals used in your study and ensure you have described the source. For more information regarding PLOS' policy on materials sharing and reporting, see https://journals.plos.org/plosone/s/materials-and-software-sharing#loc-sharing-materials.

Answer: Additional details in the method section have been added (L184-L185).

Answer: DOIs for all of the data are listed in the data availability statement together with links that reviewers can use to access the data before it is published. DOIs will be activated upon manuscript acceptance. Appropriate DOIs (but still inactive) are also listed below.

Data availability:

Figure 1.

Doi: 10.6084/m9.figshare.8234153

Figure 2.

Doi: 10.6084/m9.figshare.8234192

Figure 3.

Doi: 10.6084/m9.figshare.8234189

Figures S1 and S2

Doi: 10.6084/m9.figshare.8234156

Tables S1 and S2.

Doi: 10.6084/m9.figshare.8234159

---

## [Decision Letter · Decision Letter 1]

2 Mar 2020

(-)-α-Pinene reduces quorum sensing and Campylobacter jejuni colonization in broiler chickens

PONE-D-19-20048R1

Dear Dr. Smole Možina,

We are pleased to inform you that your manuscript has been judged scientifically suitable for publication and will be formally accepted for publication once it complies with all outstanding technical requirements.

With kind regards,

Patrick Jon Biggs, PhD

Academic Editor

PLOS ONE

Additional Editor Comments (optional):

Reviewers' comments:

Reviewer's Responses to Questions

**Comments to the Author**

1. If the authors have adequately addressed your comments raised in a previous round of review and you feel that this manuscript is now acceptable for publication, you may indicate that here to bypass the “Comments to the Author” section, enter your conflict of interest statement in the “Confidential to Editor” section, and submit your "Accept" recommendation.

Reviewer #1: All comments have been addressed

Reviewer #3: All comments have been addressed

2. Is the manuscript technically sound, and do the data support the conclusions?

Reviewer #1: Yes

Reviewer #3: Yes

3. Has the statistical analysis been performed appropriately and rigorously? 

Reviewer #1: Yes

Reviewer #3: Yes

4. Have the authors made all data underlying the findings in their manuscript fully available?

Reviewer #1: Yes

Reviewer #3: Yes

5. Is the manuscript presented in an intelligible fashion and written in standard English?

Reviewer #1: Yes

Reviewer #3: Yes

6. Review Comments to the Author

Reviewer #1: (No Response)

Reviewer #3: I reviewed the R1 as well as the responses from the authors to the former reviewers. I think the authors did a good job to address these questions. I do not have any further concern.

7. PLOS authors have the option to publish the peer review history of their article (what does this mean?). If published, this will include your full peer review and any attached files.

Reviewer #1: Yes: Marc Heyndrickx

Reviewer #3: Yes: Xiaonan Lu

---

## [Editor Report · Acceptance letter]

12 Mar 2020

PONE-D-19-20048R1 

(-)-α-Pinene reduces quorum sensing and *Campylobacter jejuni* colonization in broiler chickens

Dear Dr. Smole Možina:

I am pleased to inform you that your manuscript has been deemed suitable for publication in PLOS ONE. Congratulations! Your manuscript is now with our production department. 

With kind regards,

on behalf of

Assoc Prof Patrick Jon Biggs 

Academic Editor

PLOS ONE